# A Few Adversarial Tokens Can Break Vision Transformers

## Abstract

Vision transformers rely on self-attention operations between disjoint patches (tokens) of an input image, in contrast with standard convolutional networks. We investigate fundamental differences between the adversarial robustness properties of these two families of models when subjected to adversarial token attacks (i.e., where an adversary can modify a tiny subset of input tokens). We subject various transformer and convolutional models with token attacks of varying patch sizes. Our results show that vision transformer models are much more sensitive to token attacks than the current best convolutional models, with SWIN outperforming transformer models by up to $\sim 20\%$ in robust accuracy for single token attacks. We also show that popular vision-language models such as CLIP are even more vulnerable to token attacks. Finally, we also demonstrate that a simple architectural operation (shifted windowing), which is used by transformer variants such as SWIN, can significantly enhance robustness to token attacks. Further, using SWIN as a backbone for vision-language models improves robustness to token attacks. Our evaluation, therefore, suggests that using SWIN backbones or BEiT style pretraining results in models more robust to token attacks.

## 1 Introduction

### 1.1 Motivation

Vision transformers (Dosovitskiy et al., 2020), or ViTs, are now ubiquitous across the entire spectrum of tasks in computer vision. ViT-based models now rank among the state-of-the-art for a variety of tasks, while also providing additional benefits like zero-shot classification (Radford et al., 2021) and distributional robustness (Fang et al., 2022).

At the heart of vision transformers is the *self-attention* operation, a mechanism that allows the network to find and exploit non-local correlations between spatially-disjoint, potentially-far away patches of a given input image; indeed, an image can now be viewed as a collection of disjoint patches. In the context of vision, small non-overlapping patches serve as input *tokens* to the transformer. Models such as ViT, Data Efficient Image Transformers (DeIT) (Touvron et al., 2021), and many other variants all rely on this token-based mechanism to represent images. In comparison, convolutional networks (CNNs) take raw image pixels as input features, and each layer only calculates localized correlations.

By now, it is well-known that convolutional networks (CNNs) are vulnerable to adversarial attacks under a variety of threat models (Szegedy et al., 2013; Carlini & Wagner, 2017b; Croce & Hein, 2019). We can therefore also ask: how well do vision transformers fare against adversarial attacks? This has previously been addressed by several papers (Bhojanapalli et al., 2021; Paul & Chen, 2021) which showed that ViTs are at least as robust as CNNs under norm-bounded adversarial perturbation attacks.

However, since transformers process inputs in the form of tokens, this motivates a unique threat model, that is not captured by previously proposed norm-bounded perturbation models. In this work, we introduce the "token attack" threat model for transformer-based architectures, where a malicious attacker can modify a tiny number of tokens (imperceptibly or otherwise). Specifically, we attempt to answer the question:

*Are transformers robust to malicious perturbations to only a handful of input tokens?*

Our findings bear both good and bad news. On the negative side, we show that vanilla ViT models are worryingly brittle when subjected to token attacks: *even a single adversarially designed token in an image can dramatically affect performance*, compared to similar attacks on modern convolutional models (such as ConvNextv2). This brittleness to token attacks is *even worse* when we consider CLIP vision-language models with transformer backbones. Therefore, any pre-processing pipeline that processes images token-wise before feeding into ViTs should be handled with care from a security perspective.

However, on the positive side, we show that some *modern variations of ViTs are significantly more robust* than standard ViTs. As a potential explanation for why this is the case, we identify an architectural operation common to these models ("shifted windowing"). This may provide insights on how to design transformer-style vision architectures in the future that are robust by construction.

## 1.2 Our contributions

We introduce the "token attack" model where an adversary is permitted to modify a small subset of $K$ tokens (patches) of a given input image comprising $N$ tokens. For large $N$ and $K$, finding the optimal attack under this threat model is combinatorially hard, but we can employ natural relaxations that can be solved by (projected) gradient descent; see Section 3 below for technical details.

Using our token attack, we interrogate vulnerabilities of several families of neural architectures; transformer-based (ViT (Dosovitskiy et al., 2020), DeIT (Touvron et al., 2021), BeIT, and others), convolutional (Resnets (He et al., 2016), WideResNet Zagoruyko & Komodakis (2016), ConvNextv2), and finally vision-language models that perform zero-shot classification (specifically, CLIP (Radford et al., 2021)) with a transformer-based image backbone.

In summary, we make the following contributions:

1. We propose a saliency based token attack that leverages block-sparsity projected gradient descent.
2. With our token attack algorithm, we can significantly degrade the performance of the vision transformers using only a small number of tokens (corresponding to 0.5% of pixels) — as opposed to $\ell_2$- or $\ell_\infty$-attacks which rely on perturbing all image pixels. We show consistent degradation of classification performance of all architectures on token attacks of increasing patch sizes and number of patches.
3. We demonstrate that, for token attacks accounting for the architecture and token size, transformer architectures relying on non-overlapping patches are less robust as compared to convolutional networks. Intriguingly, we also show that CLIP (Radford et al., 2021) models based on transformer backbones, which have generally been shown to be robust to distribution shifts, are far less robust to token attacks.
4. We also observe that models that have reduced dependency on singular tokens, generally achieved through overlapping patches through shifted windowing or masked pretraining (SWIN, ConvNextv2 and BeIT) are more robust than other models. We further analyse this effect through various experiments on SWIN and show that using overlapping patches leads to robustness.

## 2 Related Work

***Vision Transformers:*** Transformers, introduced by Vaswani et al. (2017), have led to significant improvements in NLP tasks. Following this success in NLP, Dosovitskiy et al. (2020) propose Vision Transformers (ViT) that leverage non-overlapping patches as tokens input to a similar attention based architecture. ViTs have led to significant developments across vision tasks, including zero-shot classification (Radford et al., 2021), captioning (Li et al., 2022), and image generation (Rombach et al., 2021) among others. Vision transformers have further been improved through the use of distillation (Touvron et al., 2021), masked image pre-training (BeIT) (Bao et al., 2022) and linear time attention layers (Liu et al., 2021). Given the recent ubiquity of vision transformers across computer vision, it is of great importance to quantify and analyse their robustness to adversarial perturbations.

***Adversarial attacks:*** Deep networks are vulnerable to imperceptible changes to input images as defined by the $\ell_\infty$ distance (Szegedy et al., 2013). There exist several test-time attack algorithms with various threat models: $\ell_p$ constrained (Goodfellow et al., 2015; Kurakin et al., 2017; Carlini & Wagner, 2017a),

black-box (Ilyas et al., 2018b;a), geometric attacks (Engstrom et al., 2017; Xiao et al., 2018), semantic and meaningful attacks (Joshi et al., 2019; Zhang et al., 2019b; Song et al., 2018) and data poisoning based (Shafahi et al., 2018).

***Defenses:*** Due to the vast variety of attacks, adversarial defense is a non-trivial problem. Empirical defenses as proposed by Madry et al. (2018), Zhang et al. (2019a), and Jagatap et al. (2020) rely on adversarial data augmentation and modified loss functions to improve robustness. Samangouei et al. (2018) and Yin et al. (2020) propose preprocessing operations as defenses. However, such defenses fail to counter adaptive attacks (Athalye et al., 2018). Wong & Kolter (2018), Cohen et al. (2019) and Salman et al. (2019) provide methods that guarantee robustness theoretically.

***Patch attacks:*** Patch attacks (Brown et al., 2017) are a more practically realizable threat model. Zolfi et al. (2021), Thys et al. (2019), and, Wu et al. (2020) have successfully attacked detectors and classifiers with physically printed patches. In addition, Croce & Hein (2019) and Croce et al. (2020) also show that spatially limited sparse perturbations suffice to consistently reduce the accuracy of classification model. This motivates our analysis of the robustness of recently invented architectures towards sparse and patch attacks.

***Attacks and defenses for vision transformers:*** The popularity of transformer models in image classification have inspired a number of studies about their robustness. Bhojanapalli et al. (2021) and Hendrycks et al. (2020) analyse the performance of vision transformers in comparison to massive ResNets under various threat models and concur that vision transformers (ViT) are at least as robust as Resnets when pretrained with massive training datasets.

The transferability of adversarial attacks on ViT has also been examined. Mahmood et al. (2021) showed that adversarial examples do not transfer well between CNNs and transformers, and build an ensemble based approach towards adversarial defense. Naseer et al. (2021) and Wei et al. (2021) suggested that adversarial attacks can be transferred between ViTs and CNNs by specifically tailoring attacks to transformers. We consider an orthogonal setup, where we construct adversarial attacks specifically for transformer models to leverage the special input modality. Qin et al. (2021) show that ViTs are specifically vulnerable to patch-level transformations, leading to good in-distribution accuracies but poor out-of-distribution performance. Salman et al. (2022) present a certified defense for patch attacks, where in ViTs outperform Resnets.

Paul & Chen (2021) claim that ViTs are robust to a large variety of corruptions due to the attention mechanism. However, Lovisotto et al. (2022) show that dot-product attention can result in vulnerability to adversarial patch attacks and propose adversarial objectives for crafting patches that target this explicitly. Gu et al. (2021) find that ViTs are more effective in dealing with naturally distorted image patches compared to CNNs but are more susceptible to adversarial patches, where the attention mechanism can be easily fooled to focus more on the adversarially perturbed patches. Fu et al. (2022) implement a patch attack by using a set of attention-aware optimization techniques that are specifically designed to deceive the self-attention mechanism of the model.

Wang et al. (2022) show that it is possible to improve the robustness of CNNs to changes in natural distribution shifts by patchifying input images without incorporating any attention-related techniques. Croce & Hein (2022) show that the patchified stem notably improves the robustness with respect to $\ell_2$ attacks while being comparable to $\ell_\infty$ attacks. Shi & Han (2022) propose a decision-based black box attack leveraging the noise sensitivity of different subsets of an input image.

In any case, there seems to exist a strong effect on model robustness when subjected to patch-wise (token) perturbations. In this paper, we illuminate this effect in greater detail for several model families. We achieve this by leveraging a new saliency based adversarial token attack. Our approach not only validates previously observed phenomenon, but also suggests the existence of easily found *vulnerable* tokens through simple saliency metrics. In comparison with previous work (Gu et al., 2021; Qin et al., 2021; Croce & Hein, 2022), we analyse a larger variety of models, including modern variants like CLIP. We also show through ablations that techniques like shifted windowing architectures and masked patch pretraining, which reduce dependence on single patches, further provide significant robustness towards adversarial patches. Further, we study models adversarially finetuned with our proposed token attack, and observe that our approach exposes interesting connections in the transformer architecture and dependence on individual tokens. Finally, we

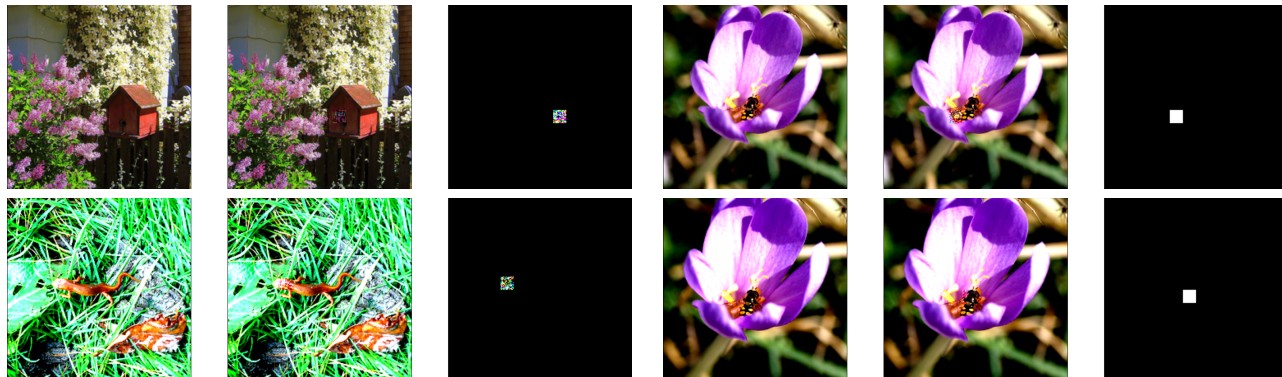

Figure 1: **Examples of token attacks.** *Token attacks are successful in creating nearly imperceptible perturbations that fool ViTs. The leftmost image in every triplet is an original image, followed by the adversarial image with a single token perturbed, and the token perturbation.*

also compare our proposed attack against the greedy attack used in Gu et al. (2021) and show that our attack is more efficient as well as better at finding adversarial tokens.

## 3    Token Attacks on Vision Transformers

We begin by introducing Token attacks, which specifically are tailored towards targeting transformer architectures that rely on patch-based inputs.

**Threat Model:** Let $\mathbf{x} \in \mathbb{R}^d$ be a $d$-dimensional image, and $f : \mathbb{R}^d \to [m]$ be a classifier that takes $\mathbf{x}$ as input and outputs one of $m$ class labels. For our attacks, we focus on sparsity as the constraining factor. Specifically, we restrict the number of pixels or blocks of pixels that an attacker is allowed to change. We consider $\mathbf{x}$ as a concatenation of $B$ blocks $[\boldsymbol{x}_1, \ldots \boldsymbol{x}_b, \ldots, \boldsymbol{x}_B]$, where each block is of size $p$. In order to construct an attack, the attacker is allowed to perturb up to $K \leq B$ such blocks for a $K$-token attack. We also assume a white-box threat model, that is, the attacker has access to the model including gradients and preprocessing. We consider a block sparse token budget, where we restrict the attacker to modifying $K$ patches or "token" with an unconstrained perturbation allowed per patch.

**Sparse attack:** We first consider the simpler case of a sparse ($\ell_0$) attack. This is a special case of the block sparse attack with block size is *one*. Numerous such attacks have been proposed in the past (Papernot et al., 2016; Wiyatno & Xu, 2018). The general idea behind most such attacks is to analyse which pixels in the input image tend to affect the output the most $S(x_i) := \left| \frac{\partial L(f(\mathbf{x}, \mathbf{y}))}{\partial x_i} \right|$, where $L(\cdot)$ is the adversarial loss, and $c$ is the true class predicted by the network. The next step is to perturb the top $s$ most salient pixels for a $s$-sparse attack by using gradient descent to create the least amount of change in the $s$ pixels to adversarially flip the label.

**Patchwise token attacks:** Instead of inspecting saliency of single pixel we check the norm of gradients of pixels belonging to non-overlapping patches using patch saliency:

$$S(\mathbf{x}_b) := \sqrt{\sum_{x_i \in \boldsymbol{x}_b} \left| \frac{\partial L(f(\mathbf{x}, \mathbf{y}))}{\partial x_i} \right|^2},$$

for all $b \in \{1, \ldots B\}$. We pick top $K$ blocks according to patch saliency. The effective sparsity is thus $s = K \cdot p$. The sequence of operations are summarized in Alg. 1.

Since the transformers we test use non-overlapping patches as tokens, we select those as input to the algorithm. Fig. 1 shows examples of token attacks on transformers.

---

**Algorithm 1** Adversarial Token Attack

---

**Require:** $\mathbf{x}_0$: Input image; $f(.)$: Classifier; $\mathbf{y}$: Original label; $K$: Patch budget; $p$: Patch size.

1: Set $i \leftarrow 0$

2: $[b_1 \ldots b_K]=$ Top-K of $S(\mathbf{x}_b) = \sqrt{\sum_{x_i \in \boldsymbol{x}_b} \left| \frac{\partial L(f(\mathbf{x},\mathbf{y}))}{\partial x_i} \right|^2}$, $\forall b$.

3: **while do** $f(\mathbf{x}) \neq y$ OR MaxIter

4: $\quad \mathbf{x}_{b_k} = \mathbf{x}_{b_k} + \nabla_{\mathbf{x}_{b_k}} L; \quad \forall \ b_k \ \in \ \{b_1, \ldots, b_K\}$

5: **end while**

---

## 4 Experiments and Results

**Setup:** To ensure a fair comparison, we choose the best models for the ImageNet dataset (Russakovsky et al., 2015) reported in the literature.

The models achieve near state-of-the-art results in terms of classification accuracy. They also are all trained using the best possible hyperparameters for each case. We use these weights and the shared models from the `Pytorch Image models` (Wightman, 2019) repository.

**Models:** In order to quantify the robustness of transformers to other architectures, we consider multiple families of models: Vision Transformers (ViT) (Dosovitskiy et al., 2020; Touvron et al., 2021; Bao et al., 2022), Resnets (He et al., 2016; Zagoruyko & Komodakis, 2016), ConvViTs (Wu et al., 2021), ConvNexts (Liu et al., 2022), SWIN (Liu et al., 2021), FlexiViT (Beyer et al., 2023) and CLIP (Radford et al., 2021). We note that the vision transformer architectures except SWIN rely on non-overlapping patches. SWIN uses a shifted window based approach to construct tokens. Note that Dosovitskiy et al. (2020) show that best performing ImageNet models have a fixed input token size of $16 \times 16$. We, therefore, fix a token size of $16 \times 16$ for all our models.

We do a hyper parameter search to find the best attacks for each model analysed. However, we use the same number of steps for all our experiments.

**Patch attacks:** We allow the attacker a fixed budget of tokens as per Algorithm 1.

We use the robust accuracy as the metric of robustness, where a higher value is better. We start with an attack budget of 1 token for an image size of $224 \times 224$ for the attacker where each token is a patch of the size $16 \times 16$. In order to compensate for the differences in the size of the input, we scale the attack budget for ConvNextv2-Huge by allowing for a bigger patch size ($24 \times 24$ to be precise) to be perturbed. For this setup, we do not enforce any imperceptibility constraints. We run the attack on the ImageNet validation set for the network architectures defined above. Fig. 2 and Fig. 3 show the result of our analysis. Notice that vision transformer architectures are less robust as compared to ResNet-101 and ConvNextv2 models. However, we observe that SWIN and BeIT reject this trend and are more robust than CNNs (including ConvNext-v2) for lower token budgets and comparable for higher budgets. We conjecture that this is a consequence of the architectural novelties that SWIN and BeIT use. SWIN, for example, leverages shifted windowing and BeIT on the other hand, uses a mask-based pretraining approach which intuitively reduces the models' dependence on a single patch. We empirically validate this conjecture in the next section by ablating over the amount of patch-overlap.

***Varying the Token budget:*** We now study the robustness of models by varying the token budget. For this case, we only study attacks for a fixed patch (token) size of $16 \times 16$. See Fig. 2 for the results. We clearly observe a difference in the behavior of transformer models and convnets here. In general, for larger token budgets, SWIN outperforms all other token based models. For smaller token budgets, while transformers are still comparably robust, convnets tend to be more robust than ViT. In addition, the robust accuracies for Transformers fall significantly for as few as *four* tokens. The advantage offered by distillation in DeIT is also lost under token attacks. In addition, a surprising observation is that CLIP models are vulnerable to even a single token attacks. This is of particular concern as CLIP embeddings are now used for a variety of downstream tasks. We also analyse finetuned CLIP models from Wortsman et al. (2022) and observe that while they improve in robustness over the zero-shot CLIP, the models are still worse than convolutional

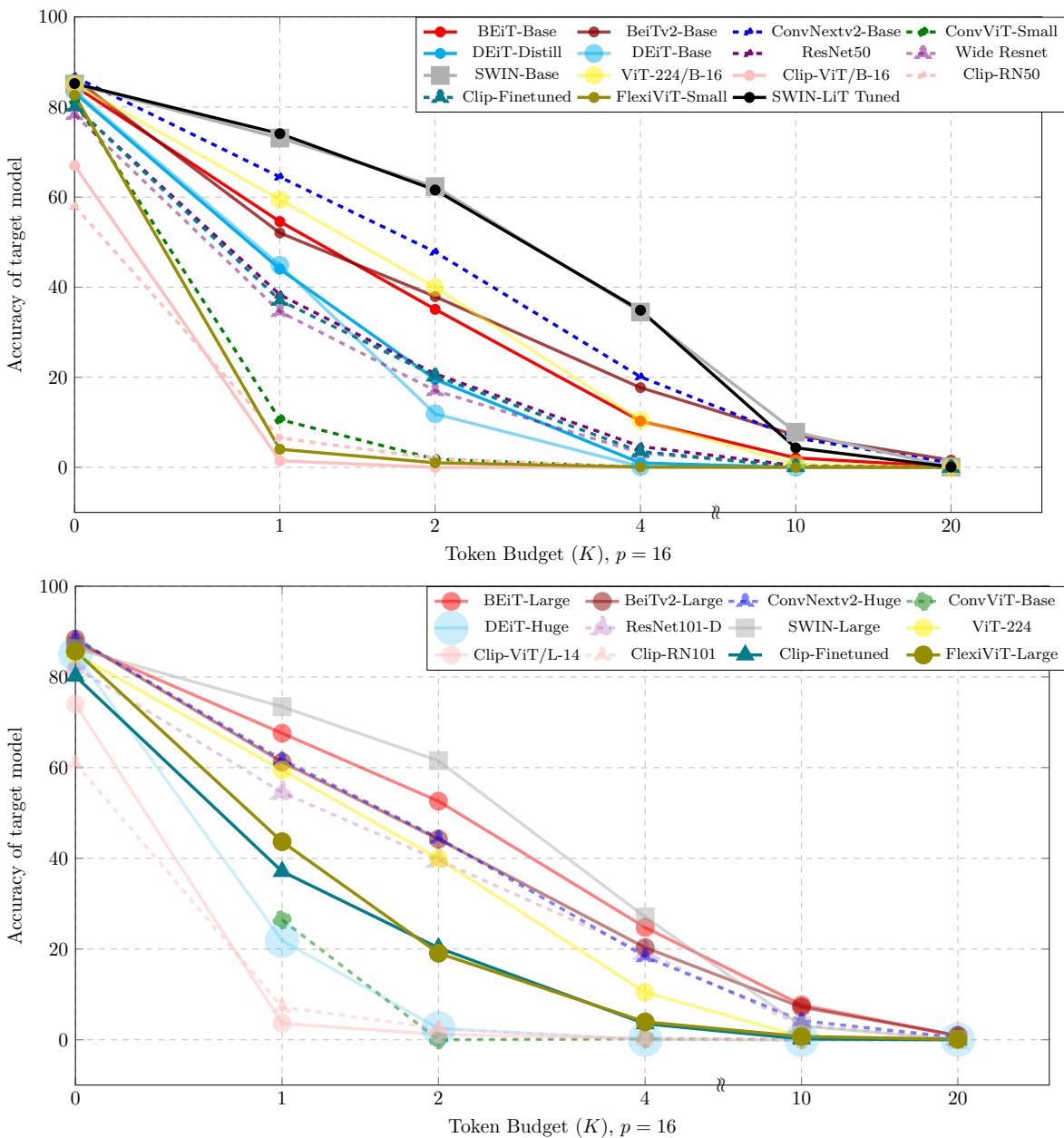

Figure 2: **Token attacks with varying budgets.** $p = 16$. *Vision transformers are less robust than SWIN, BeIT and convnets for token budgets greater than 2. Note that ViT-224/B-16 takes a 196 $16 \times 16$ tokens as input. The results are split across two figures based on approximate model sizes to ensure better visibility. Detailed results for all models can be found in the appendix.*

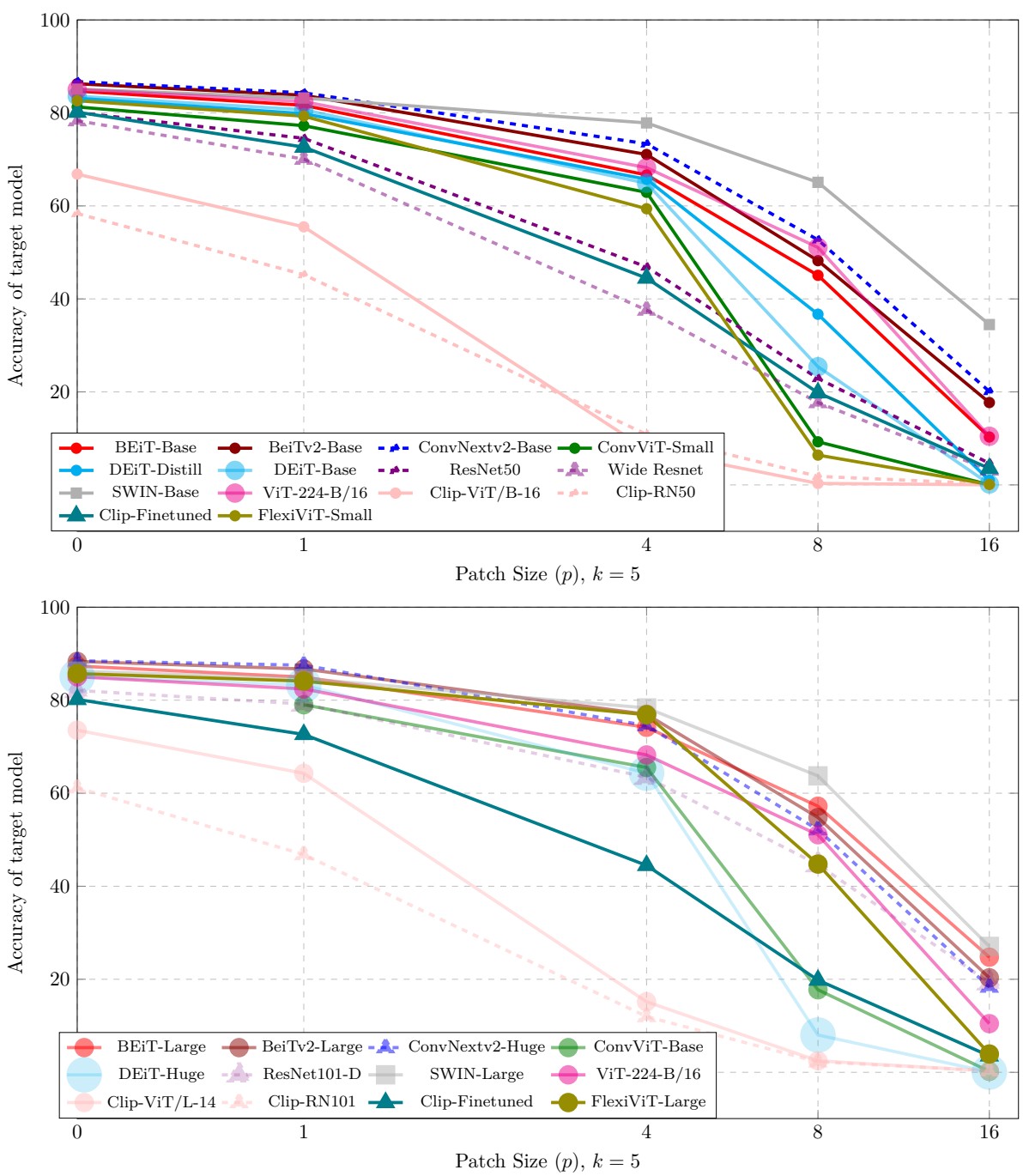

Figure 3: **Token attacks with varying patch sizes.** $K = 5$. *When the attack patch size is smaller than the token size of the architecture, all models except CLIP are comparably robust against patch attacks. However, as the attack patch size approaches token size, significant deterioration in robustness for vision transformers can be observed. The results are split across two figures based on approximate model sizes to ensure better visibility. More detailed results can be found in the Appendix.*

Table 1: **Robust accuracies,** $s = 256$ **sparse and** $K = 1$**,** $16 \times 16$ **token attack**

| Model | Clean | Sparse | Token |
|---|---|---|---|
| BEiT-Base-224 | 84.69 | 29.28 | 54.46 |
| BEiT-Large-224 | 87.34 | 42.60 | 67.58 |
| BEiTv2-Base-224 | 86.27 | 45.16 | 52.05 |
| BEiTv2-Large-224 | 88.34 | 52.03 | 61.23 |
| ConvNextv2-Base | 86.72 | 44.77 | 64.47 |
| ConvNextv2-Huge | 88.48 | 35.46 | 61.76 |
| ConvNextv2-Large | 86.89 | 51.01 | 65.45 |
| ConvViT-Base | 82.18 | 12.96 | 26.44 |
| ConvViT-Small | 81.28 | 13.61 | 10.62 |
| ConvViT-Tiny | 73.48 | 18.35 | 4.03 |
| DeiT224-Distill | 83.16 | 24.06 | 44.03 |
| DeiT3-Base-224 | 83.61 | 12.51 | 44.87 |
| DeiT3-Huge-224-14 | 85.07 | 6.76 | 21.87 |
| DeiT3-Large-224 | 84.62 | 8.58 | 55.27 |
| DeiT3-Medium-224 | 82.86 | 24.04 | 46.59 |
| DeiT3-Small-224 | 81.46 | 4.14 | 21.34 |
| FlexiViT-Small | 82.62 | 11.92 | 4.02 |
| FlexiViT-Base | 84.82 | 21.40 | 14.29 |
| FlexiViT-Large | 85.71 | 50.65 | 43.66 |
| ResNet101-D | 82.10 | 33.78 | 54.53 |
| ResNet50 | 80.10 | 9.03 | 38.33 |
| Wide Resnet | 78.33 | 4.78 | 34.59 |
| SWIN-224 | 82.90 | 48.42 | 69.11 |
| SWIN-224-Base | 85.11 | 48.65 | 73.11 |
| SWIN-224-Large | 86.24 | 48.00 | 73.43 |
| ViT-224 | 85.03 | 25.44 | 59.46 |

models. We consider two models from their setup: (1) the best performing finetuned model, and (2) the averaged greedy-soup model. We observe that the finetuned models perform better than the zero-shot CLIP models for low token budgets. However, as token budgets increase ($> 4$ tokens), the robust accuracy drops to nearly zero in both instances. However, SWIN-LiT tuning, which uses a CLIP-style contrastive vision language training approach, outperforms all other models. This suggests that the SWIN architecture is naturally robust to token level attacks.

Another intriguing observation shows that FlexiViTs, which are agnostic to patch sizes, are also equally vulnerable to token attacks. While being generally comparable to DEiT and CLIP, they lag behind ViTs and Resnets.

***Varying patch sizes:*** In order to further analyse if these results hold across stronger and weaker block sparse constraints, we further run attacks for varying patch sizes. Smaller patch sizes are equivalent to partial token manipulation. We fix the token budget to be 5 tokens. Here, this corresponds to allowing the attacker to perturb 5 $p \times p$ patches. See Fig. 3 for the results. As one would expect, a smaller partial token attack is weaker than a full token attack. Surprisingly, the Transformer networks are comparable or better than ResNets and other convnets for attack sizes smaller than a single token. This leads us to conclude that Transformers can compensate for adversarial perturbations within a token. However, as the patch size approaches the token size, SWIN, BeIT, and convnets outperform ViTs and ConvViTs. Notice that CLIP follows the same trend as well with CLIP-finetuned models being slightly more robust than the

zero-shot classifier. On the other hand, FlexiViTs, now, are more robust to sub-token attacks. Specifically, the patch size agnostic architecture allows the models to deal with sub-token attacks effectively. However, as we increase the patch size to $16 \times 16$, FlexiViTs are equally vulnerable.

***Comparison with greedy token attacks (Gu et al., 2021).*** To further understand the efficacy of our token attack, we also compare it against the greedy token attack proposed in Gu et al. (2021); Karmon et al. (2018). Firstly, note that the greedy token attack is restricted to a single token. We ensure that a fair comparison by running both attacks with 32 iterations with a token budget of one on the pretrained ViT-B/16. We record both the final robust accuracy and average time to find an adversarial token for the two attacks in Table 2. The attack times were calculated using a single A100 GPU with a batch size of 1.

Table 2: **Comparison with Greedy token attacks.** Observe that our proposed attack achieves a higher attack success rate (lower robust accuracy) and is more efficient as compared to the greedy approach adapted from Karmon et al. (2018) by Gu et al. (2021).

| Algo. | Robust Acc. | Avg. time per image (s) |
|---|---|---|
| Adv. Token Attacks (ours) | 59.46 | 1.27 |
| Gu et al. (2021) | 74.4 | 73.75 |

***Adversarial Training.*** We further study the effect of adversarial training Madry et al. (2018) on the existing models. Due to computational restrictions as well as to ensure that models are not undertrained, we rely on adversarial finetuning to train our models. For generating attack images, we use our proposed token attack with a token budget of 5 patches, and 16 iterations. We train three models: ViT-B/16-224, SWIN-B, and Resnet-101D, and use early stopping following Rice et al. (2020). We then test the robustness of our adversarially trained models using our token attacks with token budgets of 1 and 5. See Table **??** for results. Intriguingly, while adversarial training helps ViTs, it actually degrades performance in the case of SWIN, and Resnet101d, suggesting that ViTs are vulnerable due to the existence of specific 'vulnerable' tokens.

Table 3: **Adversarial Training.** Observe that the robustness of the ViT significantly increases over the base model, while SWIN and Resnet101 actually decrease, suggesting existence of specific vulnerable tokens for the ViT. The robust accuracies are reported as $a/b$ where $a$ represents the accuracy for a token budget of 1 while $b$ represents the same for 5

| Architecture | Base Clean Acc. | Base Robust Acc. | Adv. Clean Acc. | Adv. Robust Acc. |
|---|---|---|---|---|
| ViT-B/16 | 85.03 | 59.46 / 10.46 | 81.28 | 66.52 / 35.90 |
| SWIN-B | 85.11 | 73.11 / 34.48 | 84.06 | 74.32 / 33.08 |
| Resnet101D | 82.10 | 54.53 / 19.28 | 78.71 | 52.82 / 12.30 |

**Ablation Study: Saliency v/s Random Selection.** We also analyse the efficacy of using the saliency metric to select vulnerable patches. To compare, we randomly select 1, 2 or 5 tokens and run steps 2-4 from Alg. 1. Fig. 4 shows the results of the experiment. Our saliency based block-sparse attacks outperform random sampling and is able to reduce the accuracies of all vision transformer models for lower token budgets. This demonstrates the necessity of using a saliency based metric to select tokens for attack.

**Ablation Study: Sparse Attacks:** We also study the effect of the block-sparsity constraint which forces token level attacks here. The sparse variant of our algorithm restricts the patch size to $1 \times 1$. We allow for a sparsity budget of 0.5% of the original number of pixels. In the case of the standard $224 \times 224$ ImageNet image, the attacker is allowed to perturb 256 pixels. We compare the attack success rate of both sparse attack and patch-based token attack at same sparsity budget; to compare we chose $1, 16 \times 16$ patch attack (refer Table 1). Notice that sparse attacks are generally stronger as compared to token attacks. We see that as is the case with token attacks, even for sparse attacks, vision transformers are less robust as compared to ResNets. With the same sparsity budget, sparse attacks are stronger than token attacks; however we stress

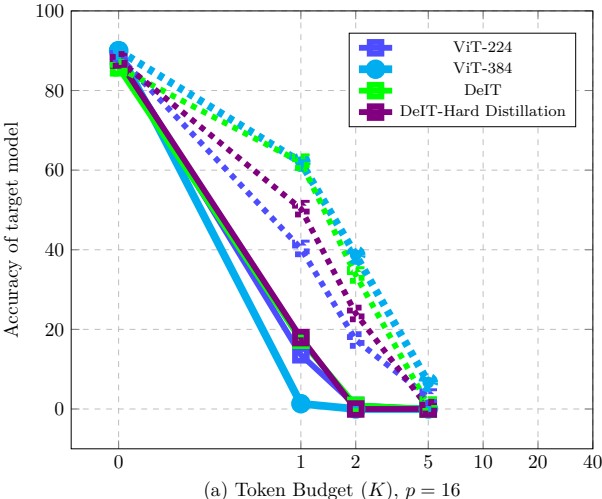

(a) Token Budget ($K$), $p = 16$

Figure 4: **Saliency based Token sampling v/s Random Sampling**: The solid lines represent robust accuracies for our token attack whereas dotted lines show the same for randomly sampled tokens. Notice that our saliency based token attack is more successful at constructing attacks with fewer tokens compared to random sampling.

that sparse threat model is less practical to implement as the sparse coefficients may be scattered anywhere in the image.

## 5 Effects of Shifted Windowing in SWIN on Token Robustness

Observing that SWIN and ConvNextv2 perform much better, we conjecture that this is because these models reduce the model dependency on single tokens. Primarily, The SWIN architecture utilizes two different window partitioning strategies in consecutive blocks. The first block applies a regular window partitioning strategy, referred to as W-MSA. In contrast, the next block uses a shifted windowing strategy, referred to as SW-MSA. In the SW-MSA approach, the window configuration is shifted relative to the previous block by an amount determined by the shift size parameter. The self-attention computation in the shifted windows crosses the boundaries of the windows in the previous layer, providing connections among them. This introduces redundancy across tokens, thus reducing dependency on singular patches.

Table 4: ***Robust Accuracy for SWIN models trained with different shift sizes.*** *Notice that the SWIN model trained with non-overlapping patches is more vulnerable to adversarial token attacks.*

| Shift (Patch Overlap) | Clean | Patch Size | | | |
|---|---|---|---|---|---|
| | | 1 | 4 | 8 | 16 |
| 0 | 81.04 | 78.56 | 71.62 | 60.31 | 33.52 |
| 1 | 82.01 | 79.75 | 73.82 | 64.00 | 36.18 |
| 2 | 82.02 | 80.18 | 75.20 | 66.80 | 42.49 |
| 3 | 81.94 | 79.89 | 74.37 | 64.55 | 38.07 |

To investigate the effectiveness of shifted windowing in SWIN, we trained SWIN transformers with varying shift sizes in the SW-MSA and analysed their robustness to patch attacks. The shift size of the model decides how many patches overlap across the window and the shifted window. We extended the model from the `Pytorch Image models` (Wightman, 2019) repository to change the number of overlapping pixels across patches in the W-MSA and SW-MSA layers. The original model uses a shift size of 3. We train three additional models with shift sizes of zero, one and two. Note that all configurations have a standard window size of 7×7. We train these models using the standard settings defined in the original paper (Liu et al., 2021). We then repeat our experiments for the three models and compare the robust accuracy.

We find that on comparing the models with shifted windowing (shift size of 1, 2, and 3) to the non-shifted window variant, the robustness increases; see Table 4. Further, we note that there is a higher difference between 0-shift to 1-shift compared to others. This clearly shows that reducing the independent, non-overlapping token dependency plays a major role in improving the robustness of the transformers to token attacks.

We also show that using the SWIN backbone as an architecture for vision-language models allows for robustness to token attacks (refer Fig. 2).

## 6  Discussion

In this paper, we analyzed the robustness of various modern vision classifiers to architecture-specific token attacks. To achieve this, we proposed a new block-sparsity based gradient attack that leverages a form of saliency to select and perturb vulnerable tokens. We then used this attack to analyse a variety of classifiers including various flavors of vision transformers. To provide a baseline, we also compared modern convolutional networks under the same attack.

Analysing the above results, we infer certain interesting properties of transformers.

1. We find that Vision Transformers are generally susceptible to token attacks, even for very low token budgets.
2. However, ViTs appear to compensate for perturbations to patch attacks smaller than the token size. This suggests that the patch size used for tokenization plays a significant role in ensuring that transformers being robust to token perturbations.
3. Intriguingly, while subsequent adversarial finetuning allows ViTs to be comparably robust to SWIN and Resnets, robustness of SWIN and Resnets actually degrades, suggesting existence of uniquely vulnerable tokens in ViTs.
4. We also observe that pure convolutional models (ResNet, ConvNextv2), as well as transformers like SWIN and BeIT are more robust to such token level attacks. Further analysis of SWIN models reveals that using shifted windowing helps reduce dependence of the model predictions on a few tokens, and improve robustness through enforcing redundancy.
5. Finally, we note that CLIP is especially vulnerable to token attacks. However, a SWIN transformer trained using a CLIP-style loss is able to subvert token attacks.

Our analysis reveals serious vulnerabilities for many modern backbone architectures. Architectural novelties like SWIN-based shifted windowing and training approaches like masked pretraining (BEiT) that reduce the dependence on singular tokens show some robustness. However, future investigation into better token and shifted windowing schemes will be instrumental in ensuring more robust and reliable models.

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

## A  Experiments

For all experiments, we use SGD for optimization with a learning rate of 0.1 for a maximum of 100 steps.

For Flexivit, which was trained on patches with patch size varying from $\{48, 40, 30, 24, 20, 16, 15, 12, 10, 8\}$, patch size of 16 was chosen (even when the image size is $240 \times 240$) because the next possible attack token size, 20, would result in very strong attack, making the results incomparable

To investigate the effectiveness of shifted windowing in Swin, we extended the Timm repository by incorporating support for new model configurations. These new configurations consisted of shift sizes of 0, 1, and 2, while the default configuration used a shift size of 3 (All configurations have a standard window size of 7). Each model was trained for 300 epochs using the adamw optimizer. To ensure consistency in comparing results, we also included the default model with shift size 3 in our experiments.

For zero-shot CLIP models such as CLIP-ViT/B-16, we utilized the Timm repository. For Finetuned Clip models, we used the model soup repository, which contains multiple models fine-tuned with various hyper-parameter configurations on ImageNet. Fine-tuning was performed end-to-end to modify all parameters, resulting in better accuracy than only training the final linear layer. Among these individual fine-tuned models (i.e., not the greedy soup), the best performing model was selected. These experiments use the CLIP ViT-B/32 model.

## B  Detailed Results

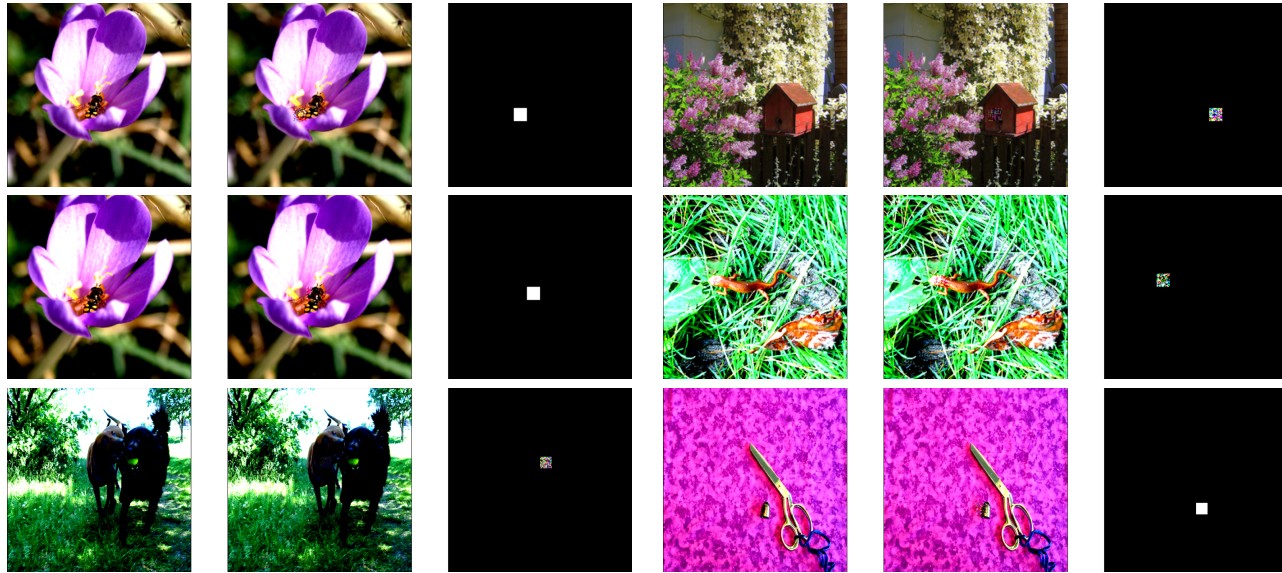

Figure 5: **Patch attacks on Transformers**: *The attack images are generated with a fixed budget of 1 patch.*

Table 5: *Robust Accuracy vs Token Budget.* The models are attacked with a $p = 16$. Note that for smaller token budgets, the models perform nearly the same. However, as the token budget increases, Convnets are more robust than Vision Transformers.

| Model | Clean | Token Budget | | | | |
|---|---|---|---|---|---|---|
| | | 1 | 2 | 5 | 10 | 20 |
| BEiT-Base-224 | 84.69 | 54.46 | 35.09 | 10.30 | 2.11 | 0.12 |
| BEiT-Large-224 | 87.34 | 67.58 | 52.61 | 24.76 | 7.67 | 0.85 |
| BEiTv2-Base-224 | 86.27 | 52.05 | 37.91 | 17.71 | 7.12 | 1.61 |
| BEiTv2-Large-224 | 88.34 | 61.23 | 44.23 | 20.32 | 7.18 | 0.95 |
| Conv2-Base | 86.72 | 64.47 | 47.81 | 20.10 | 6.54 | 0.98 |
| Conv2-Huge | 88.48 | 61.76 | 44.49 | 18.32 | 4.20 | 0.49 |
| Conv2-Large | 86.89 | 65.45 | 48.29 | 24.15 | 9.63 | 2.19 |
| ConvViT-Base | 82.18 | 26.44 | 7.72 | 0.21 | 0.02 | 0.01 |
| ConvViT-Small | 81.28 | 10.62 | 1.83 | 0.05 | 0.01 | 0.01 |
| ConvViT-Tiny | 73.48 | 4.03 | 0.24 | 0.01 | 0.01 | 0.01 |
| DeiT224-Distill | 83.16 | 44.03 | 19.54 | 0.98 | 0.01 | 0.01 |
| DeiT3-Base-224 | 83.61 | 44.87 | 11.88 | 0.20 | 0.01 | 0.01 |
| DeiT3-Huge-224-14 | 85.07 | 21.87 | 2.48 | 0.05 | 0.01 | 0.01 |
| DeiT3-Large-224 | 84.62 | 55.27 | 16.07 | 0.33 | 0.02 | 0.01 |
| DeiT3-Medium-224 | 82.86 | 46.59 | 17.61 | 0.67 | 0.01 | 0.01 |
| DeiT3-Small-224 | 81.46 | 21.34 | 4.88 | 0.25 | 0.01 | 0.01 |
| FlexiViT-Small | 82.62 | 4.02 | 1.02 | 0.09 | 0.01 | 0.01 |
| FlexiViT-Base | 84.82 | 14.29 | 4.08 | 0.40 | 0.02 | 0.01 |
| FlexiViT-Large | 85.71 | 43.66 | 19.08 | 3.93 | 0.72 | 0.10 |
| ResNet101-D | 82.10 | 54.53 | 39.62 | 19.28 | 3.17 | 0.37 |
| ResNet50 | 80.10 | 38.33 | 20.78 | 4.65 | 0.37 | 0.04 |
| Wide Resnet | 78.33 | 34.59 | 17.05 | 3.22 | 0.19 | 0.02 |
| SWIN-224 | 82.90 | 69.11 | 58.36 | 30.73 | 5.42 | 0.05 |
| SWIN-224-Base | 85.11 | 73.11 | 62.33 | 34.48 | 7.81 | 0.06 |
| SWIN-224-Large | 86.24 | 73.43 | 61.52 | 27.07 | 3.04 | 0.03 |
| ViT-224 | 85.03 | 59.46 | 39.95 | 10.46 | 0.70 | 0.01 |
| CLIP-ViT/B-16 | 66.83 | 1.48 | 0.14 | 0.02 | 0.01 | 0.01 |
| CLIP-ViT/L-14 | 73.54 | 3.61 | 1.27 | 0.28 | 0.11 | 0.04 |
| CLIP-RN50 | 58.36 | 6.55 | 1.97 | 0.23 | 0.01 | 0.01 |
| CLIP-RN101 | 61.18 | 7.24 | 2.53 | 0.33 | 0.03 | 0.01 |
| CLIP-Finetuned | 80.16 | 37.10 | 20.26 | 3.53 | 0.18 | 0.01 |
| SWIN (LiT-Tuned) | 85.12 | 74.09 | 61.59 | 34.88 | 4.34 | 0.1 |

Table 6: *Robust Accuracy vs Patch Size.* The models are attacked with a $K = 5$. Note that for smaller patch sizes, the models perform nearly the same. However, as the patch size increases, Convnets are more robust than Vision Transformers.

| Model | Clean | Patch size | | | |
|---|---|---|---|---|---|
| | | 1 | 4 | 8 | 16 |
| BEiT-Base-224 | 84.69 | 81.63 | 66.66 | 45.07 | 10.30 |
| BEiT-Large-224 | 87.34 | 84.96 | 74.17 | 57.21 | 24.76 |
| BEiTv2-Base-224 | 86.27 | 83.80 | 71.06 | 48.22 | 17.71 |
| BEiTv2-Large-224 | 88.34 | 86.71 | 76.97 | 54.77 | 20.32 |
| Conv2-Base | 86.72 | 84.31 | 73.36 | 52.71 | 20.10 |
| Conv2-Huge | 88.48 | 87.52 | 74.53 | 52.15 | 18.32 |
| Conv2-Large | 86.89 | 84.93 | 75.72 | 55.84 | 24.15 |
| ConvViT-Base | 82.18 | 79.02 | 65.53 | 17.74 | 0.21 |
| ConvViT-Small | 81.28 | 77.26 | 62.92 | 9.27 | 0.05 |
| ConvViT-Tiny | 73.48 | 68.43 | 52.32 | 6.55 | 0.01 |
| DeiT224-Distill | 83.16 | 79.84 | 65.73 | 36.70 | 0.98 |
| DeiT3-Base-224 | 83.61 | 80.68 | 64.86 | 25.42 | 0.20 |
| DeiT3-Huge-224-14 | 85.07 | 83.24 | 64.31 | 8.02 | 0.05 |
| DeiT3-Large-224 | 84.62 | 82.49 | 68.56 | 26.84 | 0.33 |
| DeiT3-Medium-224 | 82.86 | 79.60 | 68.02 | 35.38 | 0.67 |
| DeiT3-Small-224 | 81.46 | 77.00 | 46.47 | 12.62 | 0.25 |
| FlexiViT-Small | 82.62 | 79.30 | 59.39 | 6.41 | 0.09 |
| FlexiViT-Base | 84.82 | 82.81 | 62.15 | 16.11 | 0.40 |
| FlexiViT-Large | 85.71 | 84.10 | 76.86 | 44.77 | 3.93 |
| ResNet101-D | 82.10 | 79.17 | 63.44 | 44.65 | 19.28 |
| ResNet50 | 80.10 | 74.54 | 46.88 | 22.89 | 4.65 |
| Wide Resnet | 78.33 | 70.06 | 37.60 | 17.71 | 3.22 |
| SWIN-224 | 82.9 | 80.83 | 74.02 | 61.64 | 30.73 |
| SWIN-224-Base | 85.11 | 83.19 | 77.85 | 65.07 | 34.48 |
| SWIN-224-Large | 86.24 | 84.67 | 78.33 | 63.73 | 27.07 |
| ViT-224 | 85.03 | 82.42 | 68.23 | 51.07 | 10.46 |
| CLIP-ViT/B-16 | 66.83 | 55.50 | 7.62 | 0.32 | 0.02 |
| CLIP-ViT/L-14 | 73.54 | 64.28 | 15.17 | 2.47 | 0.28 |
| CLIP-RN50 | 58.36 | 45.22 | 10.87 | 1.84 | 0.23 |
| CLIP-RN101 | 61.18 | 46.82 | 11.99 | 2.24 | 0.33 |
| CLIP-Finetuned | 80.16 | 72.64 | 44.48 | 19.82 | 3.53 |

