# OpenReview forum: "A Few Adversarial Tokens Can Break Vision Transformers"
_TMLR — Rejected by TMLR_

### Review · Reviewer_FJwg · 2023-05-11

**Summary Of Contributions:**

This work investigates the vulnerability of vision transformers to adversarial attacks on individual tokens. The authors propose a saliency-based token attack algorithm that degrades the performance of vision transformers significantly using only a small number of tokens. They show that vision transformers are more sensitive to token attacks than convolutional networks and that popular vision-language model such as CLIP are even more vulnerable to token attacks. However, they also demonstrate that certain modern variations of vision transformers, such as SWIN, are more robust to token attacks due to their use of shifted windowing or masked pretraining. Overall, the paper provides insights on how to design transformer-style vision architectures that are robust by construction and highlights the need for careful consideration of pre-processing pipelines that process images token-wise from a security perspective.

**Audience:**

Yes

**Broader Impact Concerns:**

Based on the information provided in the paper, it does not appear that the authors have explicitly addressed the limitations of their work or provided an ethical statement. It is important for authors to acknowledge the potential limitations of their study and discuss any ethical considerations related to their research, including the potential impact on society or the use of the findings in a harmful manner. This is especially important in work on adversarial attacks.

**Claims And Evidence:**

Yes

**Requested Changes:**

1. Related Work & Novelty: The authors can include a more comprehensive and updated review of related work in their paper. They should clearly explain how their work is different from the existing literature and highlight the novelty of their contribution. They can also discuss how their approach complements or improves upon previous works.
2. Robust models: The authors can evaluate the robustness of their proposed method on robustified models to provide additional insight. They can consider evaluating models that are adversarially trained, or models that incorporate other robustness techniques.
3. Extended evaluation of shifted windowing: The authors can extend their evaluation of shifted windowing by studying its impact on adversarial training. They can also compare the performance of shifted windowing with other robustness techniques and explore if they can be used together to further improve the robustness of vision transformers.


**Strengths And Weaknesses:**

## Strengths
(+) Empirical results: The paper provides empirical evidence that vision transformers are more vulnerable to token attacks than convolutional networks, with SWIN outperforming transformer models by up to 20% in robust accuracy for single token attacks. The paper also shows that popular vision-language models such as CLIP are even more vulnerable to token attacks.
(+) Insights on robustness: The paper identifies an architectural operation common to modern variations of vision transformers ("shifted windowing") that enhances robustness to token attacks.
(+) Technical contributions: The paper proposes a saliency-based token attack algorithm that leverages block-sparsity projected gradient descent, which enables the degradation of the performance of vision transformers using only a small number of tokens.
(+) Clarity: The paper is well-written and organized.

## Weaknesses:
(-) Related Work & Novelty: Some related works are missing. [A, D] also set out to attack patches (tokens) and presents very similar results to this work. [B, C] also study patch attacks on vision transformers and provides a theoretical understanding of this vulnerability. Other works in the space include [E, F]. Given the above-mentioned references and the works listed in the paper, the authors do not fully differentiate themselves from these works. Hence, overall the technical novelty and the provided insight to the readers are not clear.
(-) This work mainly focuses on non-robust models. However, it is well known that ViTs are vulnerable to adversarial examples and patch attacks. I believe an evaluation of robustified models would provide additional insight.
(-) While the analysis of the shifted windowing is interesting, an extended evaluation would be interesting, if this technique can further improve adversarial training.

[A] Are Vision Transformers Robust to Patch Perturbations?; ECCV 2022
[B] Give Me Your Attention: Dot-Product Attention Considered Harmful for Adversarial Patch Robustness; CVPR 2022
[C] Decision-based Black-box Attack Against Vision Transformers via Patch-wise Adversarial Removal; NeurIPS 2022
[D] On the interplay of adversarial robustness and architecture components: patches, convolution and attention; Workshop at ICML 2022
[E] Adversarial Robustness Comparison of Vision Transformer and MLP-Mixer to CNNs; BMVC 2021
[F] On the Adversarial Robustness of Vision Transformers; TMLR 2022

---

> ### Author Response · Authors · 2023-06-01
> **Thank you for the detailed remarks**
>
> Thank you for your review and feedback! We thank you for pointing us to related work. We will update our manuscript to cite and discuss these works in greater detail. However, we will briefly discuss some differences between our work and others.
>
> Firstly, we validate findings from [A] and [D] which use greedy methods to find adversarial tokens (patches) using our new saliency based approach, which is more efficient at finding these tokens (see table above).
> [D] actually perturbs a larger number of tokens (20%), while also enforcing an additional $\ell_p$ constraint which is a slightly different threat model. On the other hand, we specifically focus on perturbations bounded to a few tokens with only a single saliency step to select vulnerable tokens to. Secondly, we also provide specific insights on models robust to such attacks by studying properties unique to SWIN. We show that the token redundancies introduced by SWIN allow for better protection against token-level changes and may be an important factor to constructing future architectures. [B] and [C] are complementary to our work with a different approach to constructing image-wide adversarial attacks leveraging dot-product attention and decision based methods to find adversarial tokens. [E] and [F] also use different threat models from our setup.
>
> We have also trained exemplar models adversarially (see table 3 in revised manuscript), and show that adversarial training using our saliency based attack leads to greater gains in robustness for ViTs over other architectures.
>
> The idea of adding SWIN-like architectural components to other adversarial defense techniques is really interesting. Given the large variety of possible approaches, we restricted our study to just understanding the effect of shifted windowing in isolation here. However, we hope to study the complex interactions of SWIN with other adversarial techniques in greater detail in future work.

---

> > ### Comment · Reviewer_FJwg · 2023-06-12
> > **Thank you for the rebuttal**
> >
> > I appreciate the authors response. My concerns were addressed and I do not have further questions at this point.

---

### Review · Reviewer_4GVj · 2023-05-17

**Summary Of Contributions:**

This work explores the effect of a new threat model: token attack on state of the art vision transformer models. The paper proposes a saliensy-based method for finding a small set of tokens to attack. Emperical results are provided showing a small number of tokens can already lead to a strong attack for vision transformer models, and such attack is more effective on ViT than on CNN models. Observations are also made on the improved robustness of SWIN transformer against the proposed attack.

**Audience:**

Yes

**Claims And Evidence:**

Yes

**Requested Changes:**

See previously mentioned weakness part. It would be great to discuss the effect if the attack patch is allowed to go across the boundary of image token.

**Strengths And Weaknesses:**

## Strengths

This is an overall well written emperical paper. The paper identifies the threat model of token attack on ViT and shows the success of proposed attack. The paper provides adequate ablation study and comparison results to support the claims made in the paper.

## Weakness

1. The attack discussed in the experiment is always aligned with the token of the attacked model, where the attacked patch always lies within a token. It would be interesting to see if the attack can be stronger when the patch can be applied to any locations in the image, not necessarily within an image token boundary.

2. For a complete evaluation of the attack, it would be good to also include attack transferablity results. It would be interesting to see if attack generated on one type of ViT cna transfer to another type of ViT model.

---

> ### Author Response · Authors · 2023-06-01
> **Thank you for the positive feedback!**
>
> Thank you for the positive feedback. We appreciate your taking the time to read our paper in great detail and providing insightful comments.
>
> We agree that studying non-grid aligned attacks is an interesting direction. Croce and Hein, 2022 study this in some detail for a simple greedy specific $\ell_p$ bounded attack and observe XCiT and Resnet-50 outperforms DeIT for  non-grid aligned attacks. Our focus here, however, was to observe if there are specific vulnerabilities that transformers incur due to tokenization. We therefore study sub-token adversarial noise and observe that sub-token attacks (for example 8x8 adversarial patches for ViT-B/16) are less effective on transformers showcasing their ability to correct for adversarial noise spread across various tokens..
>
> We thank the reviewer for suggesting transfer attacks. We agree that transfer attacks are a very interesting direction. Specifically, given the effectiveness of simple saliency measures in choosing vulnerable tokens, it is of great interest to see if these vulnerabilities are common across models.  If the reviewer is okay with it, we will add a paragraph discussing the possibilities and probe this in greater detail in another paper.

---

### Review · Reviewer_qtby · 2023-05-18

**Summary Of Contributions:**

This paper studies the robustness of ViTs against patch attacks. While previous works show that ViTs are more robust than CNNs against regular Lp ball attacks, this paper shows that ViTs are more sensitive to patch attacks compared to CNNs, and CLIP with ViT as the backbone is more vulnerable. The paper also mentions that models such as SWIN which reduce the dependency on a single token are more robust.


**Audience:**

Yes

**Claims And Evidence:**

No

**Requested Changes:**

1. I would suggest the authors discuss more on the difference between this work and the existing works on patch attacks for ViTs, as well as adding empirical comparisons wherever applicable.
2. Section 5 also needs to be revised to address weakness point #3 mentioned above.


**Strengths And Weaknesses:**

Strengths:
1. The paper shows a saliency based token attack on ViTs.
2. The paper shows that ViTs and CLIP with the ViT backbone are vulnerable to the token attack.
3. The paper also shows that some models reducing the dependency on a single token tend to be more robust to the token attack.

Weaknesses:
1. There have been several other papers studying the patch attack for ViTs (such as Qin et al., 2021; Gu et al., 2021; Fu et al., 2022) and they have also claimed that ViTs are vulnerable to patch attacks. Existing works such as Gu et al., 2021 have also discussed models such as SWIN. Thus the conclusion that ViTs are vulnerable to patch/token attacks while some variants are more robust does not seem to be quite new given the existing works.
2. While this paper claims a contribution on proposing a new attack, it is unclear how/why the attack method differs from the patch attacks in the existing works (such as Qin et al., 2021; Gu et al., 2021; Fu et al., 2022). Also, there is no empirical comparison between the proposed attack and those in the existing works.
3. In Section 5, the paper claims that "Shifted Windowing in SWIN helps", and "as the shift size increases from zero, the robustness increases; see Table 2". While it looks true when the shift size increases to 1 or 2, the claim doesn't seem to hold when the shift size increases to 3, according to Table 2. Thus the claims in Section 5 sound questionable to me.

---

> ### Author Response · Authors · 2023-06-01
> **Thank you for the insightful remarks**
>
> We thank the reviewer for their detailed and insightful comments. We agree that there have been other works which observe similar token-level vulnerabilities in transformers. However, we differ from existing works in two specific areas. Firstly, we leverage a new token attack mechanism that relies on saliency to select vulnerable tokens. We compare with Gu et al. who leverage a greedy token attack, and find that our approach achieves better robust accuracy more efficiently. We will add more detailed comparisons in the final draft.
>
> | Algo | Model | Robust Acc. |  Avg. Time per image (s) |
> | --- | --- | --- | --- |
> | Adv. token attacks | ViT-B/16 | 59.46 | 1.27  |
> | Gu et al. | ViT-B/16 | 74.4 | 73.75 |
>
> Further, in comparison with other related works, we study a larger variety of models (including vision-language models) and show that our adversarial token attacks help in analyzing vulnerabilities at the token level. Specifically, even the simple saliency measure we used allows for selecting vulnerable tokens. We also discuss some additional points of difference in our comment to Reviewer FJwg. We appreciate this discussion, and have edited our related work and will add experimental comparisons to include these changes.
>
> For Sec. 5, we specifically would like to point out that while we do not observe monotonic behavior, we do observe improvement over 0-shift SWIN, thus suggesting a link between the shifted window operation and token-level robustness. However, we agree that the statement may be construed differently, and have rephrased the section to avoid confusion. We hope to study the specific reasons for non-monotonic behavior in detail in future work, though we conjecture this effect occurs due to corner effects.

---

> ### Comment · Reviewer_qtby · 2023-06-11
> **Post-rebuttal feedback**
>
> Thanks to the authors for the response.
>
> I think the authors are claming using "saliency to select vulnerable tokens" as a contribution compared to the prior works. However, in Fu et al., 2022, they have already mentioned that a saliency-based method was worse than their "attention-aware patch selection" (see Table 1 in Fu et al., 2022).
>
> Overall, given the multiple existing works, it is unclear how the proposed saliency-based attack differs from the one considered in Fu et al., 2022, and this paper would need a revision to include comprehensive experiments to compare the proposed method with the existing works (not only with Gu et al., 2021 on a single setting).

---

### Author Response · Authors · 2023-06-01
**Thank you for the detailed reviews. Addressing some common observations**

We thank the reviewers for their insightful comments. We are encouraged that all reviewers find (1) strong empirical evidence of adversarial tokens on vision transformer performance (2) technical novelty of saliency based token attacks, (3) the quality of writing, to be strengths of our paper. We focus on the main concerns raised by the reviewers - a more detailed discussion of similar papers, and adversarial training as a defense. While we will address the individual comments, broadly, our paper proposes a “saliency based” token attack to identify vulnerable tokens in vision transformer based models. We leverage this attack to both validate previous work, as well as extend the studies to a larger group of models including now pervasive vision-language models. Further, we also study mechanisms like SWIN that exploit token redundancy to defend against token attacks.

We also thank the reviewers for suggesting improvements to our paper. In particular, as suggested by Reviewer FJwg, we present results of adversarial finetuning on Imagenet-1k with our adversarial token attack. We restrict ourselves to fine-tuning as most models we analyze have been pre-trained on larger datasets, and further fine-tuned on Imagenet-1k. We observe an interesting phenomenon, where adversarially trained ViTs are more robust than other architectures. Furthermore, we actually see a drop in robust accuracy for SWIN. This suggests that saliency based token attacks target specifically vulnerable tokens in ViTs. We have edited the manuscript to include these results (see Tab. 3 in the revised draft).

| Architecture | Base Clean Acc. | Base robust Acc. | AT Clean acc. | AT robust acc. |
| ---  | ---  | --- | --- | --- |
| ViT-B16/224 | 85.03 | 10.46 | 81.28 | 35.90 |
| SWIN-B | 85.11 | 34.48 | 84.06 | 33.08  |
| Resnet-101D | 82.10 | 19.28 | 78.71 | 12.30 |

---

### Decision · Action_Editors · 2023-06-17

**Recommendation:** Reject

**Comment:**

At this stage, I find the paper is not ready to publish. The following list two main points:

1. There are some points mentioned by some reviewers, however, the rebuttal simply mentioned that the issues will be addressed but did not seem to provide enough evidence. For example, I agree with one reviewer on the transferbility of the proposed attack, which the authors simply says it will be interesting future work. I think this suggestion is valid and can definitely strengthen the paper. I highly suggest the authors to conduct such an experiment.

2. There are many missing related works. The authors add some discussions on the revision, however, I think this is not enough. Especially, one reviewer pointed out the lack of comparison of the saliency attack with Fu et al., 2022, which claimed a better method than saliency attack. The authors are encouraged to do more complete investigation on this.

Overall, the findings of the paper are interesting, however, there needs more experiments and discussions to differentiate the proposed method with existing related works. If the problems raised by the reviewers are well addressed, the paper can be a good one to accept. I encourage the authors to revise the paper accordingly, and I am happy to over see another round of reviewing process if needed.

**Audience:**

Researchers in adversarial robustness and the transformer architecture may find this paper interesting.

**Claims And Evidence:**

The claims made in the submission are indeed interesting. However, as pointed out by the reviewers, they are not fully supported by convincing and clear evidence, which are detailed in the Comment section.

**Resubmission Of Major Revision:**

The authors may consider submitting a major revision at a later time.